# A Pilot Study on Clinician-AI Collaboration in Diagnosing Depression from Speech

Kexin Feng
*Department of Computer Science and Engineering*
*Texas A&M University*
College Station, Texas, USA
kexin@tamu.edu

Theodora Chaspari
*Institute of Cognitive Science, Department of Computer Science*
*University of Colorado Boulder*
Boulder, Colorado, USA
theodora.chaspari@colorado.edu

*Abstract*—**This study investigates clinicians' perceptions and attitudes toward an assistive artificial intelligence (AI) system that employs a speech-based explainable ML algorithm for detecting depression. The AI system detects depression from vowel-based spectrotemporal variations of speech and generates explanations through explainable AI (XAI) methods. It further provides decisions and explanations at various temporal granularities, including utterance groups, individual utterances, and within each utterance. A small-scale user study was conducted to evaluate users' perceived usability of the system, trust in the system, and perceptions of design factors associated with several elements of the system. Quantitative and qualitative analysis of the collected data indicates both positive and negative aspects that influence clinicians' perception toward the AI. Results from quantitative analysis indicate that providing more AI explanations enhances user trust but also increases system complexity. Qualitative analysis indicates the potential of integrating such systems into the current diagnostic and screening workflow, but also highlights existing limitations including clinicians' reduced familiarity with AI/ML systems and the need for user-friendly and intuitive visualizations of speech information.**

*Index Terms*—**Human-AI Collaboration, Depression Diagnosis, Speech, Clinical AI, Decision Support Systems**

## I. Introduction

Depression, a pervasive mental health (MH) disorder, stands as a significant global concern, affecting individuals across diverse demographics and cultures. The profound impact of depression extends beyond the individual, influencing relationships, work, and overall societal well-being [1]. Depression diagnosis and screening typically relies on self-reported surveys like the Personal Health Questionnaire Depression Scale (PHQ-8) [2] or clinical interviews that assess an individual's alignment with diagnostic criteria such as the American Psychiatric Association's Diagnostic Statistical Manual of Mental Disorders, Fifth Edition (DSM-5) [3]. Despite the clear criteria, healthcare professionals may still misdiagnose patients [4]. For example, female patients can be wrongfully diagnosed with depression 30-50% of the time [5]. Due to the shortage of MH professionals and the limited experience of primary care clinicians in screening for depression, numerous individuals can also go undiagnosed and lack proper screening [6]. Additionally, emotionally engaging with patients during therapy can lead to burnout among therapists themselves [7].

Research in computational linguistics and speech processing

This work was supported by the National Science Foundation (CAREER: Enabling Trustworthy Speech Technologies for Mental Health Care: From Speech Anonymization to Fair Human-centered Machine Intelligence, #2046118, PI: Chaspari). The code is available at https://github.com/HUBBS-Lab/speech-depression-ensemble-learning

indicates that depression can influence psychomotor control affecting the phonological loop [8]. This is manifested in changes in speech articulation and prosody [9]. Individuals with depression often exhibit slowed speech tempo, reduced vocal pitch variability, and a tendency toward more monotonous or muted tones compared to individuals without depression [10]. Individuals with depression also depict noticeable variation in speech vowels, including shorter vowel duration with reduced variance [11] and reduced vowel space [12]. These alterations in speech can be objectively measured through acoustic properties such as prosody and timbre. By applying speech measures as an input to machine learning (ML) methods, researchers have designed various AI models to automatically identify depression from speech [13]–[15]. However, these models are rarely utilized by health professionals for various reasons, notably the limited accessibility to such models and the reluctance to rely on opaque diagnostic systems. Health professionals are typically not familiar with the extent and nature of information embedded in AI models, posing challenges in reliably interpreting the model outputs and often increasing the risk for incorrect decisions.

Here, we examine clinician's attitudes toward an assistive artificial intelligence (AI) system that employs a speech-based explainable ML algorithm for detecting depression. We describe the explainable AI system that detects depression from vowel-based spectrotemporal variations of speech, and its explanations generated through explainable AI (XAI) methods (i.e., GradCam). The system provides decisions and explanations at various temporal granularities, including utterance groups, individual utterances, and within each utterance. We present a small-scale user study with 10 participants who interacted with the system by observing its decisions and various types of explanations. We assess users' perceived usability of the system, trust in the system, and perceptions of design factors associated with several elements of the system. Quantitative and qualitative analysis of the collected data indicates the potential of integrating such systems into the current diagnostic and screening workflow, but also highlights existing limitations including clinicians' reduced familiarity with AI/ML systems and the need for user-friendly and intuitive visualizations of speech information. In summary, the contribution of this study includes:

- We designed and developed a prototype of an interactive interface that facilitates the use of a speech-based AI

model to assist with depression diagnosis.

- While prior research has focused on images or text, we acquired explanations at various levels of granularity for speech-based models which are more difficult for humans to grasp compared to visual and textual data.
- To the best of our knowledge, this is the first study exploring the interaction between health professionals and speech models for clinical diagnosis via user study.

## II. RELATED WORK

### A. Speech-based ML Models for Detecting Depression

Acoustic characteristics have long been recognized as significant indicators of depression. Research indicates that individuals with psychological and neurological disorders such as depression often exhibit a limited range of vocal frequency [16]–[18]. These differences in vocal frequency can be visually observed in spectrograms, that are characterized by extreme spectrotemporal patterns (e.g., low or high energy prevalence) and less sustained energy between frequencies [17] for patients with depression. These characteristics manifest in the speech spectrogram as irregular and often limited spectrotemporal variations. Similar effects are observed at the vowel-level, where studies indicate that individuals with depression depict a reduced frequency range between the first and second vowel formant compared to their healthy counterparts [12]. Acoustic features extracted at the vowel-level have been shown to outperform turn-level acoustic features (i.e., extracted from the entire speech turn) in identifying depression [19].

Since speech carries important information about depression, researchers have developed various ML models for automatically detecting depression from speech, with many focusing on deep learning techniques. Ma *et al.* introduced DepAudioNet, an end-to-end system that employs a 1-dimensional convolutional neural network (CNN) to encode short-term temporal and spectral correlations, followed by a long short-term memory (LSTM) network to capture long-term correlations over speech frames [13]. Similarly, Lin et al. applied a 1D CNN to the speech spectrogram to model interactions within the frequency bands, and then utilized a bidirectional long short-term memory (BiLSTM) neural network [20]. Sardari *et al.* employed a convolutional autoencoder to extract depression-based embeddings using the raw audio signal [21]. Recognizing the significant impact of speaker identity on the acoustic properties of speech, Dumpala et al. combined speaker embeddings—extracted from models pre-trained on speaker identification using a large sample from the general population—with commonly used prosodic, speech production, and spectrotemporal measures to detect depression [22] Feng & Chaspari employed a 2D CNN trained on vowel classification followed by a spatial pyramid pooling (SPP) layer that combines the vowel-based embeddings of various lengths into a final decision [23]. The SPP layer enables the model to generate explanations at different temporal granularities.

### B. Explainable ML Models of Depression Detection

While recent work has shown the effectiveness of deep learning models for depression detection, many of these models lack explainability, which is crucial for their integration into the clinical workflow. A few recent studies have started addressing the challenge of developing XAI models to estimate mental health and related outcomes from multimodal data. A review of these efforts can be found in [24]. Zogan et al., used a hierarchical attention network that applied a two-level attention mechanism at the tweet-level and word-level to calculate the importance of each tweet and each word in estimating depression from social media posts [25]. Explainability of the system was qualitatively assessed by inspecting the words and tweets that depicted the highest importance and via a word cloud that depicted the frequency of the most important words associated with five major depression symptoms (i.e., ideation, worthlessness, energy, insomnia, depression mood). Farruque et al. proposed a hierarchical mechanism that produced a text-based semantic explanation of the relevance of a tweet to the depression outcome [26]. The generated explanation was assessed in terms of its brevity and relevance to depression symptoms. Rather than using conventional word frequency approaches (e.g., bag-of-words), they represented text through a one-hot encoding process based on features that reflected potential depression symptoms, as pre-defined by medical and psychological experts. This approach enhanced the system's explainability, as the features identified as most important for the decision outcome had direct associations with depression symptoms. Kumar et al. trained an attention-based gated recurrent unit on spectral features for emotion recognition and assessed explainability of the trained network via assessing inter-emotion separability of the learned embeddings [27].

### C. User Study on Healthcare ML Models

Despite the growing interest in healthcare AI applications, few studies have explored clinicians' perceptions and attitudes toward AI models designed to assist with the healthcare continuum. Wysocki *et al.* conducted interviews with 23 health professionals regarding a Lasso regression and a random forest to aid in evaluating Coronavirus disease 2019 patient risk [28], indicating that ML was viewed positively as a tool to assist diagnosing ambiguous cases and supporting inexperienced health professionals. Rong *et al.* investigated user trust in an AI model for chest disease diagnosis using X-Ray images [29] and found that system explainability improved user trust and willingness. Recent studies have also investigated the potential of employing language models to generate explanations grounded in prior literature [30], or have solely concentrated on interviews without engaging with particular models [31]. It's important to note that previous research mainly focused on medical imaging, text, or electronic health records comprised of time series or tabular data [32], [33]. To our best knowledge, this is the first time that a user study related to AI explainability is conducted on a speech-based ML model.

## III. ASSISTIVE AI SYSTEM DESIGN

### A. AI Model Design

Inspired by the effectiveness of vowel space in discerning depression [12], prior work has proposed an AI model using vowel-related information from speech [23], which was used in this user study. This speech model offers several unique advantages, such as decent performance, providing explanations

at varying temporal granularities, and a structure designed to enhance explainability. To extract vowel information, the system comprises an encoder CNN that learns to distinguish six vowel labels (/a/, /e/, /i/, /o/, /u/, or not a vowel). The system also has a context model, which uses the embeddings of speech utterances from the encoder to identify depression.

The training of the encoder model used as an input the speech spectrograms sampled from the training set over 250ms analysis windows, with labels provided by FAVE aligner [34]. The model follows the same structure, training process, and a dynamic sampling-based data augmentation approach to address the imbalance in vowel frequency inherent to English (e.g., /a/ is more frequent than /u/) as in [23].

One layer in the encoder model, known as Spatial Pyramid Pooling (SPP) layer [35], allowed a fixed-size embedding at the output even with arbitrary-sized input, which significantly simplified the training of the context model. The context model took the embeddings of 21 utterances as input and outputted a binary decision indicating whether this utterance set showed signs of depression. We used the same model structure, training parameters, and data augmentation with perturbation to oversample the depression-labeled samples.

For testing speakers, we obtained model predictions for every 21 utterances without overlap and performed soft voting to get final decisions. This model achieved a macro-F1 score of 0.65 on the development set of DAIC-WoZ, which was a decent performance compared to other methods [23]. The sensitivity score is 0.8, comparable to multiple instruments designed for assisting diagnosis [36].

### B. AI Model Explainability
#### 1) Identify key Utterances
A clinical interview or consultation session typically lasts 45 minutes to 1 hour. The identification of pivotal utterances within the interview enables users to directly listen to AI-identified significant parts of the conversation or encourages users to pay close attention when these specific segments are played. To achieve this, we employed the Grad-Cam method that uses the context model (Section III-A) to break down the input utterance group into smaller segments, thereby showing the contribution of each utterance to the local predictions.

#### 2) Identify key parts in an utterance
Previous studies on audio visualization have primarily concentrated on enhancing the expression of emotions in speech, particularly for individuals with hearing impairments [37]–[39]. This has often involved manipulating a character's font size, spacing, height, or color to intuitively represent the pace and tone of speech. However, the identification of depression from speech is a more complicated task, and previous methods of font manipulation may not offer sufficient granularity. As a result, we opted to incorporate spectrograms into the webpage since they offer insights at the utterance level. Spectrograms exhibiting darker colors in low-energy regions and lighter colors in high-energy regions, alongside inconsistent energy levels across frequencies, may be linked to depression [17]. Since we anticipated that participants would not be familiar with the basic concepts of a spectrogram, we explicitly out-

lined these cues in the tutorial video and provided specific examples.

Beyond the raw spectrogram, we also highlighted in the interface the informative sections identified by the GradCam. Users can observe AI-identified informative sections in the spectrogram, and find possible extreme dark or light colors, or inconsistent energies as evidence for depression. The high-lighting of the spectrogram is obtained using the GradCam, and shows the most influential zones to the output of the encoder model with white highlights (Figure 1).

### C. User Interface Design and Implementation
The user interface, as depicted in Figure 1, includes various components. At the upper section of the interface, three information buttons are available, allowing users to access and review technical concepts and page-related information. Directly below these buttons, a drop-down menu allows users to select the patient ID of interest. Furthermore, users could control the audio playback using the audio controller. The AI-model probability output is located above the audio controller, facilitating the observation of model output changes while audio content is being played. The audio subtitles are on the side, allowing the user to review the conversation content quickly. The timestamps adjacent to each line enable users to easily navigate and adjust the audio playback to their preferred segments of interest. We also provide the contribution of each utterance to the model's assessment of depression. Each point in the plot is clickable, allowing users to navigate to corresponding sections of the audio if they identify segments with significant contributions to the model's depression output. Finally, at the bottom of the page, we provide the raw and highlighted parts of the spectrogram obtained in Section III-B2. These figures are segmented into syllables, providing a more granular breakdown of the information for improved comprehension. This system is constructed with Python as the backend and React as the frontend. It is hosted on a local machine and ngrok is used to enable external users (participants) to view and interact with the system on their own devices during the study.

## IV. USER STUDY WITH HEALTH PROFESSIONALS
### A. Survey Methods
#### 1) Participants
Participants were recruited using [Anonymous] university-wide bulk emails. Eligible participants were graduate students majoring in clinical psychology or related fields. We recruited 10 participants who were all female and aged between 20–35 years. This gender bias is likely due to the gender imbalance in the counseling therapy field [40]. Out of the 10 participants, 9 are doctoral students (except P10), highlighting their expertise in topics related to depression. The ten participants are evenly distributed among the three experimental conditions. We provide the demographic information and the participant assignments to each condition in Table I.

#### 2) Survey Design and Protocol
We designed a between-subject study protocol to understand the usability of assistive components on the system and its impact on user's trust and task performance. Each participant was randomly assigned to one out of three experimental

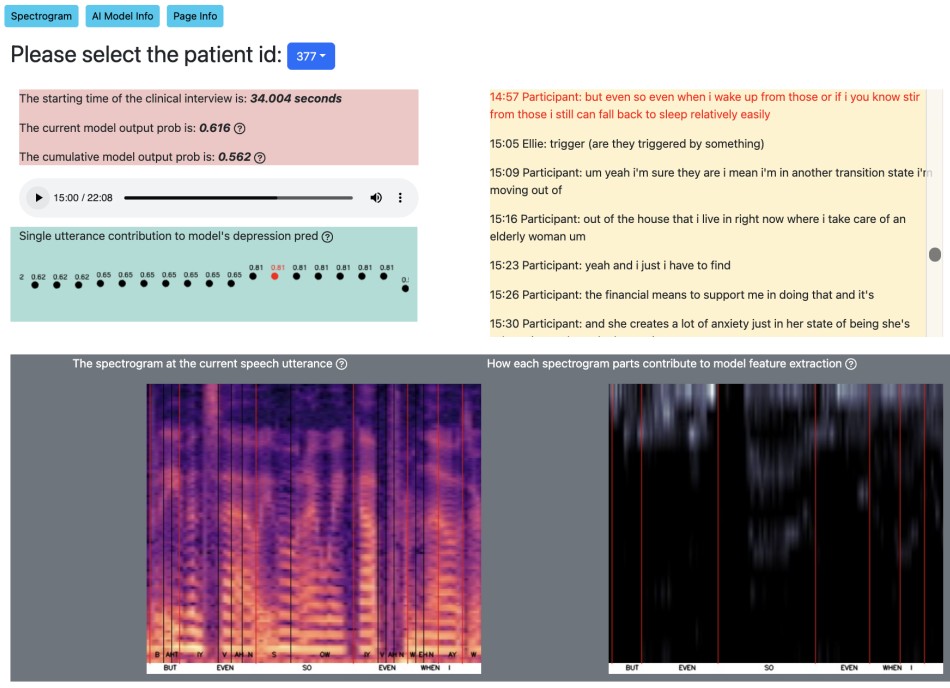

Fig. 1: A visualization of our designed interface for speech AI depression identification model.

TABLE I: The demographic info of each participant and their experimental conditions.

| ID | P1 | P2 | P3 | P4 | P5 | P6 | P7 | P8 | P9 | P10 |
|---|---|---|---|---|---|---|---|---|---|---|
| Age | 28 | 26 | 24 | 27 | 29 | 35 | 26 | 24 | 24 | 21 |
| Major | Cognitive Neuroscience | Clinical Psychology | Clinical Psychology | Clinical Psychology | Clinical Psychology | Clinical Psychology | Public Health | Education Psychology | Clinical Psychology | Public Health |
| Race | Caucasian | Asian | Black | Black | Multi-racial | Caucasian | Hispanic | Asian | Black | Asian |
| Condition | 3 | 2 | 1 | 3 | 2 | 1 | 3 | 3 | 2 | 1 |

conditions including user interface layouts featuring varying degrees of AI assistance. At the beginning of the experiment, we used a pre-recorded tutorial video associated with his or her experimental condition so that participants could familiarize themselves with technical concepts (e.g., spectrogram) and the interface. Then each participant reviewed four audio samples that were presented in one of the following three conditions: (Condition 1) The participant listened to an audio from a clinical interview, while also viewing the corresponding subtitles; (Condition 2) The participant had access to the same information as in Condition 1, and in addition, they were provided the probability of depression that the AI estimates; (Condition 3) The participant had access to the same information as in Condition 2, and in addition, they were provided with the importance that each utterance had on the decision provided by the AI, and have access to the spectrogram of the audio (i.e., visual representation of the audio) and the regions of the spectrogram that contributed the most to the AI decision.

After consenting to the study and watching a tutorial video on the system components, four audio samples were assessed by each participant. The audio samples were selected from the development set of DAIC-WOZ, and were recorded in a setting simulating real clinical interviews. The interviewer was an embodied conversational agent controlled by a human asking pre-defined questions to all participants and providing appropriate reactions to their responses. These samples have self-reported PHQ-8 scores of 3, 7, 12, and 16, representing various degrees of depression. The model's output aligns consistently with these PHQ-8 scores. The participants reviewed the four audio samples sequentially, and the order of the audio was randomized between participants. After reviewing each audio, we asked participants to answer survey questions about their decision, confidence level, workload, and trust in the AI model (if applicable). This is referred to as **between-audio survey (Table III)**. After reviewing all four audio samples, participants answered a more detailed survey related to system usability, interface design (if applicable), trust in the AI model (if applicable), and open-ended questions for their feedback. We used the System Usability Scale (SUS) survey to evaluate the user-friendliness [41], the Merit Scale survey to capture participants' trust in AI [42], and a set of customized questions to evaluate the interface design. This set of questions is known as **post-survey (Table IVa, IVb)**. SUS and Merit Scale (trust) survey questions (Table IVa) are rated on a Likert scale of 1 to 5, ranging from 'strongly disagree' to 'strongly agree'. Table IVb presents the additional questions assessing users' opinions on each interface component, using a Likert scale of 1 to 7 ranging from 'strongly disagree' to 'strongly agree'. Survey questions are not shown if they do not apply to the current experimental condition. We also included **open-ended questions (Table V)** related to the interface design that sought more detailed feedback. Finally, to check the participant's understanding of interface components, we designed a set of **knowledge-check questions (Table II)**.

TABLE II: Description of knowledge-check questions. Only applicable questions are used for each experimental condition.

| Question | Content |
| --- | --- |
| 1 | Did you have any confusion regarding the website layout? |
| 2 | What is your understanding of a spectrogram? |
| 3 | We provide two types of probabilities (local and global). What is your understanding of their difference? |
| 4 | What is the contribution of each speech utterance to the final AI decision? How was this represented in the Plot section? |

TABLE III: Description of between-audio questions. Only applicable questions are used for each experimental condition.

| Question | | Content |
| --- | --- | --- |
| Open-ended | | Do you think this person has depression? Please describe the reason. |
| Decision Confidence | 1 | I feel confident about my decision in the previous question. |
| Webpage Usage | 2 | This interface was useful for my task. |
| Workload | 3 | The task was mentally demanding. |
| Trust | 4 | I trusted the system. |

After we ran the experiments with a few participants (Phase 1), we decided to administer the knowledge-check questions right after the tutorial, to facilitate the understanding of our designed interface (Phase 2). The finalized flowchart is in Figure 2. P1 to P6 are in Phase 1 of the study, and the rest are in Phase 2.

*3) Analysis*
To analyze the quantitative data, we calculate the average score of all participants per condition. For the qualitative data, we applied the iterative coding methodology [43]. The first author (who also conducted the study) did an initial coding of the open-ended interview questions, then all the authors discussed and refined the codebook, and applied to all the responses.

*B. Quantitative Findings*
We provide the average scores per experimental condition for the between-audio and post-surveys in Figure 3. Regarding the between-audio surveys, participants in condition 3 reported the lowest workload (Between Q3), indicating effective assistance from the AI during the diagnosis task. Participants in condition 3 also depicted decision confidence comparable to that of condition 1 (Between Q1), with both surpassing condition 2. This suggested that partial AI explanations might reduce confidence in decision-making, while more comprehensive explanations contributed to reestablishing user confidence.

Following that, we analyze the results for the post-surveys in terms of the usability aspect. As we progress from condition 1 to condition 3, more components were introduced into the system. These components, being outside the users' domain knowledge, could potentially harm the website's usability. Consequently, it is natural to observe that condition 1 is the easiest to use, condition 3 exhibits the worst usability, and condition 2 falls in between. This is supported by Between Q4 and most SUS questions. However, there are some exceptions. For SUS Q1, SUS Q5, and Design Q1, condition 3 has a higher average score than condition 2. One possible explanation is that the AI decision, without any explanation, is considered less reliable for participants, which could even become a distraction if it contradicts participants' own assessments. This explanation is supported by the open-ended feedback by the participants in condition 2. For example, participant P9 said, *"If there's kind of just more details to the numbers, then I would feel a little bit more comfortable."* This explanation is also supported by Design Q6, where participants in condition 3

reported a higher score for the usefulness of the AI output. For the usability of each single component, most questions (Design Q5 to Q10) are on the higher end of the survey except Q9. This suggests that a more detailed explanation of the spectrogram should be provided to users during tutorial sessions.

Regarding the Merit Scale (Trust) survey, participants in condition 1 were excluded from this survey, because they did not interact with any AI model. We observed that participants in condition 3, despite reporting worse usability (specifically in system complexity), expressed higher trust in the system. However, they also reported poorer system consistency (Q5 and Q6). As condition 3 includes multiple AI components that could potentially conflict with each other, it is reasonable to observe higher inconsistency scores. In the between-audio survey related to user trust (Between Q4), condition 2 shows a higher score than condition 3. This indicates that the user needs time to build trust in the AI system, especially for a seemingly complex interface out of their domain.

*C. Qualitative Findings*
Participants shared their worries about using a tool that was unfamiliar to them. They talked about what they expected from a speech AI model and described how they felt while working with AI on the diagnosis task. They also discussed other possible uses for the designed interface besides diagnosis. In this section, we will discuss the findings that are important for health professionals but ignored by us computer scientists.

*1) Addressing Knowledge Gap Between Fields.*
In the healthcare field, transparency holds paramount importance. Consequently, we have strived to provide comprehensive details regarding the AI model design and decision-making processes for participants, especially in condition 3. However, we underestimated the **domain knowledge gap** across fields. P9 said, *"I felt like you're spitting out new things in that (tutorial) video, you're providing new information to somebody who is not familiar with the realm of a spectrogram and these prediction models."* Participants also characterized our system as *"advanced"* and expressed a desire for more *"laymen's terms"* in the tutorial. Moreover, participants articulated the **need for more practice** to enhance their proficiency with the system. For instance, P7 mentioned, *"the definition (of spectrogram) is clear to me, but it was not useful because I would need more practice with the inner workings to truly comprehend what it is telling me."* P8 also said, *"I don't really like looking to the spectrogram. I feel like that one I need so much knowledge to understand that."*

*2) Enhancing AI Model Design.*
Participants also shared their expectations for an ideal AI system. Several participants hoped the AI model to **be more personalized**. For instance, P7 mentioned, *"So if people have a different cadence when they talk, it may be difficult to diagnose them until the AI learns their speech patterns."* A similar recommendation was made by P1. Another participant, P5, believes that the AI model should **avoid potential biases**, particularly regional biases: *"I think also a lot of those people are in California, and so thinking regionally that's a very different kind of what that (speech) looks like."* Participants also emphasized the need for the AI model to have a **shorter**

TABLE IV: Complete System Usability Scale (SUS), Merit Scale, and interface design survey.

(a) System Usability Scale (SUS) and Merit Scale survey

| Question | | Content |
|---|---|---|
| System Usability Scale | 1 | I think that I would like to use this system frequently. |
| | 2 | I found the system unnecessarily complex. |
| | 3 | I thought the system was easy to use. |
| | 4 | I think that I would need the support of a technical person to be able to use this system. |
| | 5 | I found the various functions in this system were well integrated. |
| | 6 | I thought there was too much inconsistency in this system. |
| | 7 | I would imagine that most people would learn to use this system very quickly. |
| | 8 | I found the system very cumbersome to use. |
| | 9 | I felt very confident using the system. |
| | 10 | I needed to learn a lot of things before I could get going with this system. |
| Merit Scale | 1 | I believe the system is a competent performer. |
| | 2 | I trust the system. |
| | 3 | I have confidence in the advice given by the system. |
| | 4 | I can depend on the system. |
| | 5 | I can rely on the system to behave in consistent ways. |
| | 6 | I can rely on the system to do its best every time I take its advice. |

(b) Interface Design survey

| Question | | Content |
|---|---|---|
| Webpage Design | 1 | I was satisfied with the webpage. |
| | 2 | The contents were properly organized. |
| | 3 | The Subtitle section was useful. |
| | 4 | I was overall satisfied with the AI system. |
| | 5 | The AI probability output was easy to understand. |
| | 6 | The AI probability output was useful. |
| | 7 | The Plot section was easy to understand. |
| | 8 | The Plot section was useful in identifying key utterances. |
| | 9 | The Spectrogram section was easy to understand. |
| | 10 | The Spectrogram section was useful in providing an intuitive visualization of the audio. |

TABLE V: Description of open-ended questions. Only applicable questions are shown for each experimental condition.

| Question | Content |
|---|---|
| 1 | Did you have any confusion regarding the website layout? Are there any parts of the website that you would change? How do you think this website may assist in providing patient diagnosis? |
| 2 | What did you like the most and the least about the website? |
| 3 | What do you think about the explainability of the AI (e.g., provide reasons why model making such prediction)? How would you improve it? |
| 4 | Where can you find the definition of spectrogram in the web interface? Does the definition look clear to you? Was the spectrogram useful? |
| 5 | Will you consider using this website for assisting with diagnostic purposes? Are there any other purposes for which you might use this website? |

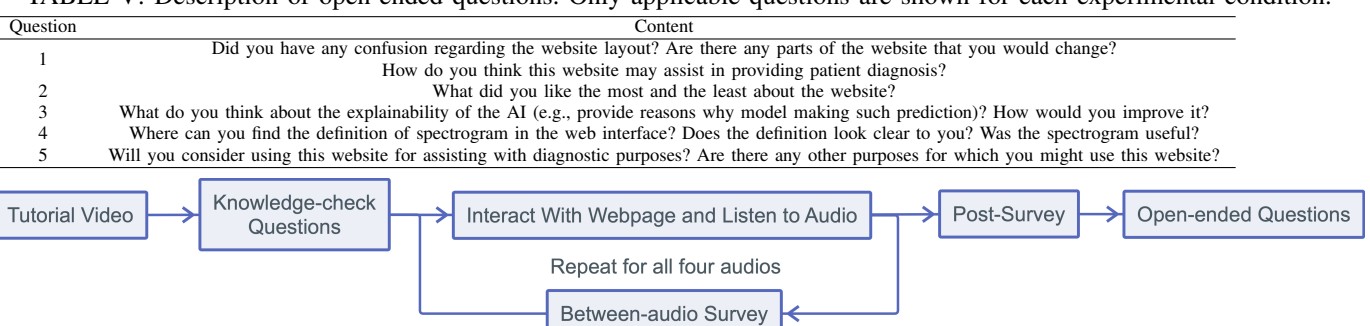

Fig. 2: A flowchart demonstrating the study protocol: interact with the system, listen to audio, and make decisions on depression.

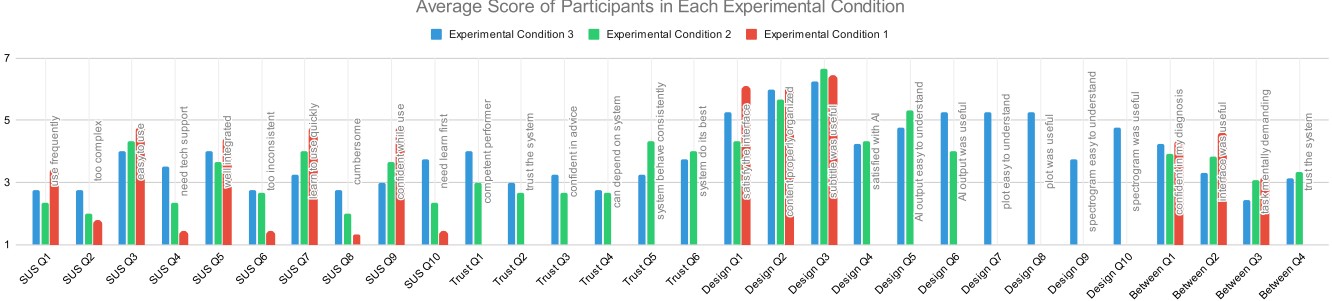

Fig. 3: Average score of System Usability Scale ('SUS'), Merit Scale ('Trust'), Interface design survey ('Design'), and Between-audio survey ('Between') between conditions. 'Trust' survey does not apply to condition 1, and some 'Design' questions do not apply to conditions 1 and 2, as relevant components are not included on the interface. The surveys are on different scales and not directly comparable. Summary of each question is provided, a higher value means more agree with the question summary.

**decision window**, providing a clearer indication of the probability of depression at specific points in the conversation: *"I feel this (current depression probability) is not changing fast enough. So sometimes it was confusing to understand like what these, to what area is referring to, like what parts of the conversation."* Lastly, participants raised concerns about the reliability of acoustic characteristics and hoped that the model could **consider other contextual factors** beyond depression. *3) Human-AI Collaboration Perspectives.*
Participants had mixed feelings during their collaboration with the AI model through our designed interface. P8 noted occasional instances where the **AI decisions seemed counter-intuitive**, stating, *"when people are discussing something positive, and then at that point, the AI is indicating the depression*

*probability is like 0.65, which is above 0.5."* However, she also acknowledged that the **AI is a competent performer**, saying, *"I feel like some part of the website is kind of accurate."* P2 also thinks the AI can be a competent performer, as she believes the AI can perceive acoustic details that are hard to capture by humans. The **lack of AI-related education** became another factor leading participants to hesitate in relying on the system. P7, for instance, mentioned, *"I wouldn't solely rely on it as much as I would rely on my skills and what I have learned (DSM-5)."* Conversely, participants also expressed appreciation for the system's **complementary role to current diagnostic criteria**: *"I think I would use it to supplement the diagnostic criteria based on what this person is telling me. I*

*would use this system or this website as a quantitative supplement,"* as stated by P9. In summary, whether participants chose to reference the AI system's output or not, their primary goal was often to **prioritize perceived decision confidence**. For example, P4 mentioned, *"Even though there's the prediction of this person probably having it (depression), I still would have to confirm, even if it's giving me this probability, I would still have to confirm based on content that meets DSM-5 criteria."*
*4) Diverse Applications Beyond Diagnosis.*
Participants actively brainstormed sessions exploring various potential applications of the AI system or the interface. Several participants highlighted its relevance in the context of **screening**. For instance, P4 said, *"It would be helpful in terms of screening and also assisting non-mental health professionals in screening, enabling them to refer individuals for follow-up. It would catch a lot of people that you were unsure of."* Participants also identified **training** as a potential application, particularly for inexperienced health professionals. P7 explicitly mentioned that the interface is well-suited for visual learners. Additionally, P5 expressed interest in utilizing the AI system for **intake** sessions, aiming for a more comprehensive initial evaluation. Moreover, participants envisioned the interface being used for **information management**. This could include providing a pre and post-treatment comparison (P7) or serving as convenient notes for easy reference (P3).

## V. SUMMARY AND FUTURE WORK

In this paper, we presented an interactive system prototype that incorporates an AI model for diagnosing depression with various explanations integrated to the AI model. We conducted a user study with health professionals using three experimental conditions: a baseline condition with no AI involvement (condition 1), AI decision included (condition 2), and explanations added (condition 3). Participants were instructed to interact with the system, listen to four audio clips, and give the diagnosis. Our findings indicate that in comparison to other types of healthcare AI systems, speech-based models are less familiar to the public, resulting in a larger knowledge gap when providing model explanations. This, on the one hand, increased the perceived complexity of the system among participants. On the other hand, it highlighted a new perspective that speech-based models could bring to the current diagnostic system. Additionally, our findings can be helpful in the design of other healthcare speech-based models beyond depression such as Parkinson's disease [44]. Our findings suggest that participants who interacted with the AI system that included the spectrogram information (condition 3) reported the lowest workload, indicating the effectiveness of AI system for helping participants make decisions. Participants in condition 3 also depicted decision confidence and trust comparable to that of condition 1, with both surpassing condition 2, suggesting that the partial explanations introduced in Condition 2 hamper participant confidence in their decision and trust in the system, as compared to providing additional explanations. However, condition 3 exhibited the worst usability and consistency across the system components, which could due to the complexity of the condition and the lack of participant knowledge

on the speech spectrogram. These indicate the feasibility of using AI systems with complex components for augmenting clinician decision-making. However, they also underscore the importance of training clinicians to better understand the systems and allowing them to practice more with these systems.

One limitation of our study is the relatively small participants size. In anticipation of the challenges associated with participant recruitment, we had initially considered a within-subject design protocol. However, there is an issue of order and carry-over effects between experimental conditions [45]. For instance, a participant might exhibit a positive reaction to condition 3 simply because it appears superior to condition 1. Additionally, participants may perceive the components presented in condition 2 as more useful than those introduced in condition 3, because they are less familiar with the spectrogram components of condition 3. Given the incremental nature of our experimental conditions (i.e., components in condition 1 is a subset of components in condition 2, components in condition 2 is a subset of components in condition 3), traditional mitigation methods like counterbalancing do not suit our study. Second, we must consider the practice effect. For instance, as participants become more adept at the depression diagnosis task, they may report survey scores indicating decreased mental demand over the course of the study [46]. While this paper presents preliminary findings from a small set of participants, we are actively recruiting participants. Our future studies will entail a more comprehensive analysis of a larger participant pool. Another limitation of the study was that due to the small sample size, we were not able to conduct a statistical analysis of the quantitative results to explore statistically significant differences among conditions. We will perform this as part of our future analysis. Finally, part of our future work should will consider ways to simplify and provide supportive clarification on the information presented in the spectrogram. This could be implemented via appropriate visualizations that could entail emphasizing the parts of the spectrogram that were deemed as the most important by the AI algorithm and providing additional explanations on spectrotemporal variations for each analysis window. An alternative approach is to provide counterfactual explanations on the spectrograms that compare and contrast different analysis windows and potentially different patients. To minimize confusion caused by the spectrogram and ensure users have an objective understanding of the interface components, we will offer more comprehensive tutorials, collaborate with the clinicians to co-design tutorials, explore improved visualization methods for speech, and provide an option to hide the spectrogram feature.

## VI. ACKNOWLEDGMENTS

We thank Dr. Adela C. Timmons, Dr. Winfred Arthur, Jr., and Dr. Brian R.W. Baucom for discussions and suggestions regarding our user study design. Our appreciation goes out to Jacqueline Duong and Sierra Walters for their feedback as pilot participants. Special thanks to Xingling Xu and Keqing Jiao for their input on UI design. We also acknowledge Zhekai Dong for discussions on the front and back-end implementation. ChatGPT was used for grammatical/syntactic edits.

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
