# OpenReview forum: "A Pilot Study on Clinician-AI Collaboration in Diagnosing Depression from Speech"
_IEEE.org/EMBS/BHI/2024/Conference — IEEE BHI'24_

### Official Review · Reviewer_vNCB · 2024-07-26
**Revision for paper 54. Good work with some points for improvement**

**Overall Rating:** 7
**Confidence:** 4

**Other Quality Metrics:**

(a) Clarity of writing: gret
(b) Clinical Significance: poor
(c) Methodological Novelty: great
(d) Experiments and Results: good

**Questions For The Authors:**

No further questions

**Strengths:**

This study explores XAI within the context of speech analysis for depression diagnosis.
The work presents a user study with health professionals to evaluate the system's usability, trustworthiness, and design. This user-centred approach ensures that the system's development is aligned with the needs and expectations of its intended users, which is very interesting.
The study emphasises the collaborative potential between clinicians and AI.

**Summary Of The Paper:**

This study investigates how clinicians perceive and interact with an AI system designed to diagnose depression. The system analyses vowel-based spectrotemporal variations in speech to detect signs of depression and employs explainable AI (XAI) methods to provide insights into its decision-making process.

**Weaknesses:**

Missing reference number 5 in the test appears as the symbol “?”
Please include a recap at the end of the introduction.
As far as I understood, participants were students with enough knowledge but no professionals with daily work experience; this fact might affect results.
The study relies on XAI methods like Grad-CAM to provide explanations. However, interpreting these explanations, particularly those related to spectrograms, can be subjective and require a certain level of familiarity with speech analysis techniques. The study acknowledges clinicians may not be comfortable interpreting spectrograms, highlighting a potential barrier to the system's adoption.

---

### Official Review · Reviewer_TuCT · 2024-07-26
**A Pilot Study on Clinician-AI Collaboration in Diagnosing Depression from Speech**

**Overall Rating:** 8
**Confidence:** 5

**Other Quality Metrics:**

(a) Clarity of writing : great
(b) Clinical Significance : great
(c) Methodological Novelty : good
(d) Experiments and Results : good

**Questions For The Authors:**

- How was the 21 utterrances selected ?
- In which context were them recorded ?
- Does only four audio used ?

**Strengths:**

Integration of AI in clinical practice is a main topic as a gap is observed between the amount of proposed system and the ones really integrated and use. This preliminary study is important and open the discussion of Clinician-AI collaboration.

**Summary Of The Paper:**

In this paper, authors aim to evaluate the human interaction with an AI system providing probability of depression from speech. In total, ten participants were involved in 3 different experiments: 1) without AI support, 2) with output probabilites from the AI model 3) with output probabilities, importance of each utterance and spectrogram. Survey questionning System usability, trust and design was conducted after using the tool for depression diagnosis on four audios. Results show positive and negative feedbacks. Provinding more explanation seems to enhance trust. More explanation made the system more complex. People consider that they could use the tool in addition to their own assessment in particular, but not by relying directly on the tool.

**Weaknesses:**

- complexity is due to the spectrogram visualization. Spectrograms were added and interpretation was made by authors (according to general knowledge on depression) and not according to the functionning of the chosen AI-model. I missed the justification for the link between the two.
- justification of using this AI-model in particular
- it would also have been interested to quanfity AI-influence in decision making
See Reverberi, C., Rigon, T., Solari, A., Hassan, C., Cherubini, P., & Cherubini, A. (2022). Experimental evidence of effective human–AI collaboration in medical decision-making. Scientific reports, 12(1), 14952.
That proposed new metric for this purpose. It may be relevant for further experiments planned by authors.
-small number of participant and small number of audio
- missing elements about the audio sessions (specially outputed probabilities and real diagnosis)

---

### Official Review · Reviewer_25Bz · 2024-07-31
**This pilot study explores the collaboration between clinicians and an AI system that employs a speech-based explainable machine learning algorithm to diagnose depression. By analyzing vocal attributes, the AI system aids in identifying depressive symptoms, aiming to enhance the accuracy and efficiency of clinical assessments.**

**Overall Rating:** 7
**Confidence:** 4

**Other Quality Metrics:**

(a) Clarity of writing – good
(b) Clinical Significance - great
(c) Methodological Novelty - great
(d) Experiments and Results - good

**Questions For The Authors:**

Introduction section: replace ? by the correct reference and Tempo should be replaced by time.

**Strengths:**

The rise of mental health issues globally has prompted the development of innovative technologies to aid in early detection and intervention. Among these, assistive artificial intelligence (AI) systems have emerged as promising tools. One such advancement is the deployment of a speech-based explainable machine learning (ML) algorithm designed specifically for detecting depression. This sophisticated system harnesses the nuances of human speech patterns to identify potential indicators of depression, providing a valuable resource for mental health professionals and individuals alike.

Advantages: By continuously monitoring speech, the system can detect early signs of depression, prompting timely intervention. Additionally, the speech-based approach is less invasive compared to traditional methods, encouraging more individuals to seek help without the stigma associated with mental.

**Summary Of The Paper:**

The authors propose an assistive AI system that employs a speech-based explainable ML algorithm for detecting depression. To the knowledge of the authors, this is the first study that a user study related to AI explainability is conducted on a speech-based ML model.

**Weaknesses:**

As reported by some of the participants, the study is a bit complicated. It will certainly make a greater contribution in a more advanced stage of research.

---

### Decision · Program_Chairs · 2024-09-23

Accept